
# Impact of the revisit frequency on cloud climatology for CALIPSO, EarthCARE, Aeolus, and ICESat-2 satellite lidar missions

Andrzej Z. Kotarba[1]

[1]Space Research Centre, Polish Academy of Sciences, 00-716 Warsaw, Poland

*Correspondence to*: A.Z. Kotarba (akotarba@abk.waw.pl)

**Abstract.** Space profiling lidars offer a unique insight into cloud properties in Earth's atmosphere, and are considered the most reliable source of total (column-integrated) cloud amount (CA), and true (geometrical) cloud top height (CTH). However, lidar-based cloud climatologies suffer from infrequent sampling: every $n$-days, and only along the ground track. This study therefore evaluated four lidar missions, namely CALIPSO (revisit every $n$=16 days), EarthCARE ($n$=25), Aeolus 10    ($n$=7), and ICESat-2 ($n$=91), to test the hypothesis that each mission provides accurate data on CA and CTH. CA/ CTH values for a hypothetical daily revisit mission were used as reference (data simulated with Meteosat 15-minute cloud observations, assumed to be a proxy for ground truth). Our results demonstrated that this hypothesis is invalid, unless individual lidar transects are averaged over an area 10×10° in longitude and latitude (or larger). If this is not the case, the required accuracy of 1 % (for CA) or 150 m (for CTH) cannot be met, either for a single-year annual or monthly mean, or for 15    a >10-year climatology. A CALIPSO-focused test demonstrated that the annual mean CA estimate is very sensitive to infrequent sampling, and that this factor alone can result in 14 % or 7 % average uncertainty with 1° or 2.5° resolution data, respectively. Consequently, applications that use gridded lidar data should consider calculating confidence intervals, or a similar measure of uncertainty. Our results suggest that CALIPSO, and its follow-on mission EarthCARE, are very likely to produce consistent cloud records despite the difference in sampling frequency.

## 1 Introduction

Accurate knowledge of cloud properties is essential for reliable modelling of the atmosphere, including climate processes (Stephens 2005; Trenberth et al. 2009). Among the many techniques used to assess cloud presence and parametrisation, satellite remote sensing plays an essential role (Stephens and Kummerow 2007). However, like other climate datasets, satellite-based climatologies face limitations related to spatial and/or temporal sampling regimes.

The coverage and frequency of satellite observations are determined primarily by the satellite's orbit, and then by the design of the cloud-sensing instrument. For a typical atmospheric mission, daily global coverage is obtained with a platform that operates from a sun-synchronous orbit: namely, a near-circular path, which is inclined at ~98° to the equatorial plane, at an altitude of up to 1000 km above Earth's surface (Capderou 2005). A sun-synchronous orbit can be designed in such a way that the satellite's ground-track repeats precisely every $n$ days: in other words, every $n$ days a satellite will pass over the



same location, and observe the land/ ocean/ atmosphere under exactly the same viewing geometry as *n* days before. The length of the revisit period (expressed in days, or number of revolutions) is constant for a mission, but may differ between missions, depending on the scientific goal (Tab.1).

**Table 1. Ground track layout parameters for the sun-synchronous lidar missions investigated in our study. EQT refers to the**
**equatorial crossing time for the ascending node, and is given in mean local solar time (LST). The full names of the lidar instruments are: ATLAS: the Advanced Topographic Laser Altimeter System; CALIOP: the Cloud-aerosol Lidar with Orthogonal Polarization; ATLID: the Atmospheric Lidar; ALADIN: the Atmospheric Laser Doppler Instrument.**

| Mission name | Lidar instrument | Ground track repeated every | | Ground tracks separation | | EQT (LST) |
|---|---|---|---|---|---|---|
| | | *n* days | orbits | km | deg. | |
| Aeolus | ALADIN | 7 | 111 | 360.6 | 3.24 | 18:00 |
| CALIPSO | CALIOP | 16 | 233 | 171.8 | 1.55 | 13:30 |
| EarthCARE | ATLID | 25 | 389 | 102.9 | 0.93 | 14:00 |
| ICESat-2 | ATLAS | 91 | 1387 | 28.8 | 0.26 | |

A less-frequent revisit schedule produces a denser ground track pattern, as the distance between adjacent tracks becomes
shorter. If a sensor's cross-track field of view (FOV) is larger than the distance between ground tracks at the equator, a single polar-orbiting mission has the ability to collect data for every single location on Earth, daily. However, when the FOV is narrower, the situation becomes challenging. Extremely narrow FOVs are specific to cloud profiling lidar sensors. Unlike imagers, profiling instruments do not focus on the spatial distribution of clouds, but on their vertical structure. To obtain vertically-resolved information, lidars emit pulses almost directly towards the nadir. The resulting swath is less than 1 km
wide, therefore obtaining global coverage is technically impossible: most locations are never sampled, while others are probed only once every *n* days (twice if ascending and descending parts of an orbit intersect).

Despite this narrow FOV, lidars are of great importance for cloud climatology. These active instruments operate very efficiently during both day and night, unlike imagers that tend to underestimate night-time cloud amount, or fail to provide cloud optical properties when no solar illumination is available (Vaughan et al. 2009; Liu et al. 2010). Moreover, lidars can
detect atmospheric features based on the time delay of the backscattered signal, which makes a direct calculation of cloud top geometrical height possible (Holz et al. 2009). Vertically-resolved information from lidars helps to unambiguously discriminate clouds and aerosols from the background, especially for locations where the spectral/ thermal contrast between cloud and the background is low (Liu et al. 2009). Finally, lidars are much more sensitive to optically-thin clouds than imagers, therefore lidar-based climatologies of cirrus, or column-integrated total cloud amount datasets are considered to be
the most reliable (Mace and Zhang 2014; Nazaryan et al. 2008).

The main source of lidar-based cloud data is the Cloud-Aerosol Lidar and Infrared Pathfinder Satellite Observation (CALIPSO) mission, equipped with the Cloud-Aerosol Lidar with Orthogonal Polarisation (CALIOP) (Winker et al. 2003). Launched in 2006, CALIPSO has been profiling the atmosphere with a revisit cycle of *n* = 16 days (233 orbits; Tab. 1). At the time of writing (the beginning of 2022), the mission has already been extended, and is approaching the end of its lifetime.





It will be followed by the EarthCARE mission ($n$ = 25 days), which will launch no earlier than 2023 (Illingworth et al.
2015). Two other lidar satellites that have operated in space simultaneously with CALIOP are Aeolus (Stoffelen et al. 2005),
and ICESat-2 (Markus et al. 2017). The former features a Doppler wind lidar, while the latter focuses on ice altimetry and
vegetation. Both missions are capable of providing limited information on clouds, but are not optimised for that goal
(Flamant et al. 2008; Palm et al. 2021).

Lidars have also been deployed on non-sun-synchronous orbits. Examples include the Cloud-Aerosol Transport System
(CATS; active between 2015–2017; Pauly et al. 2019), the Global Ecosystem Dynamics Investigation (GEDI; active since
2018; Tang et al. 2019), or the Multi-footprint Observation Lidar and Imager (MOLI; planned for 2022; Daisuke et al. 2020),
which have all been designed as host payloads for the International Space Station. The inclination of the Station's orbit
restricts observations to latitudes between 51.6°N/S, while its non-synchronous nature makes it possible to detect diurnal

cloud cycles (Noel et al. 2018) – which cannot be achieved by polar-orbiting lidar missions that operate along a fixed ground
track.

Lidar profiles from CALIPSO and other missions are explored in their native geometry (swath data, Level 2); however
climate-oriented applications frequently use gridded (Level 3) global datasets (Chepfer et al. 2010). When gridded,
individual transects are aggregated into mean monthly/ seasonal/ annual values, and averaged over predefined grid boxes.

Practical considerations mean that the grid resolution should correspond to the separation between adjacent ground tracks at
the equator, resulting in a gap-free global map. For instance, the separation for CALIPSO is 1.55° (or 172 km; Tab. 1);
therefore, global gridded datasets from the mission are usually generated at 2° resolution or coarser (e.g. Chepfer et al. 2010;
Ma et al. 2013; Kodama et al. 2012).

In gridded form, lidar observations serve primarily as ground truth for validating atmospheric models (e.g. Chepfer et al.

2008; Kodama et al. 2012; Konsta et al. 2016), or cloud climatologies from other sensors (e.g. Wylie et al. 2007; Boudala
and Milbrandt 2021; Ackerman et al. 2008; Liu et al. 2010), and they can also be analysed as independent, stand-alone cloud
climatologies (e.g. Mace et al. 2009; Adhikari et al. 2012; Oreopoulos et al. 2017). Applications that rely on gridded data can
accommodate the sparse and infrequent lidar sampling regime. Specifically, it is routinely assumed that an $n$-day revisit
schedule is sufficient to provide an accurate and reliable estimate of cloud parameters for a predefined location (grid box),

and time frame (monthly to annual average). However, this assumption has never been validated.

Lidar sampling schemes become even more significant when data from an $n$-day revisit are compared to cloud climatologies
originating from imagers with an effective 1-day schedule. In the absence of a detailed analysis, it is impossible to identify
which of the differences between datasets can be explained by inconsistent sampling schemes. Past studies (e. g. Liu et al.
2012; Kotarba and Solecki 2021) have demonstrated that the sample size of lidar (or radar) transects impacts the uncertainty

range for mean cloud amount, and appropriate confidence intervals have been suggested. Nevertheless, these studies have
not addressed the key question: how closely does $n$-day-revisit data actually approximate a 1-day revisit (imager-like)
climatology? This question motivated the present study.





Therefore, this study investigates the hypothesis that cloud parameters estimated from an *n*-day revisit mission do not differ significantly from values that would be obtained with a daily revisit. The hypothesis is tested for monthly/ annual means of
cloud amount (CA), and cloud top height (CTH), explored over a wide range of grid resolutions (1–10° latitude/ longitude). Since there are no lidar missions with a 1-day revisit schedule, the necessary data were simulated with high-temporal-resolution observations from the Meteosat satellites. Special attention is paid to CALIOP, and other sun-synchronous lidar missions are also considered for reference.

## 2 Data and methods

Sun-synchronous missions always cross the equator at the same local solar time (LST), while the latter corresponds to a different Universal Time Coordinated (UTC) hour for each transect. In order to simulate a 1-day revisit as closely as possible, lidar ground tracks must be linked with appropriate (in terms of UTC) cloud observations. Additionally, it is important that the source of cloud data is the same for all missions of interest, thus, observations must be recorded at a high temporal cadence. This requirement was met by the Meteosat satellites.

The Meteosat series are in geostationary orbit (0°E). Each satellite is equipped with the Spinning Enhanced Visible and InfraRed Imager (SEVIRI), that scans Earth's disc every 15 minutes at 1 km per pixel spatial resolution at nadir. SEVIRI's radiances are processed into a number of geophysical products. This study uses the CLoud property dAtAset using SEVIRI (CLAAS, version 2; Stengel et al. 2014; Benas et al. 2017). All CLAAS data files were accessed from the EUMETSAT Satellite Application Facility on Climate Monitoring (CM SAF).

Two parameters were investigated: cloud amount (CA; expressed on a 0–100 % scale), calculated from the CLAAS Cloud Mask product; and cloud top height (CTH; in meters), based on the Cloud Top Properties product. These parameters were chosen as they are the most accurately-reported by space lidars (Winker et al. 2017). Although lidars also provide high-quality estimations of cloud optical thickness (COT), the latter parameter was not included in the study, since Meteosat is not able to estimate it at night (unlike lidars). CA and CTH are available both day and night, both from lidars and imagers.

In this study, although Meteosat observations are considered as reference, they should not be interpreted as ground truth for CA/ CTH. Like other missions, Meteosat has limitations regarding both cloud detection and parametrization (e.g. Benas et al. 2017). It should therefore be noted that the reason for using Meteosat was not to provide absolute values of CA/ CTH, but rather to create a time series of very realistic representations of CA/ CTH at high spatial and temporal resolution.

The simulation considered the following sun-synchronous lidar missions:

— The CALIPSO mission was launched in 2006, and it is the most important source of lidar-based, long-term cloud data. The satellite, which is designed for cloud and aerosol studies, operates at two wavelengths (532 /1064 nm), with an along track laser pointing 3° off-nadir (initially 0.3°; Hunt et al. 2009). Between 2006 and 2018, the satellite flew in formation with other satellites in NASA's A-Train Constellation, contributing a unique dataset of multi-sensor collocated measurements. It remains operational, in a drifting orbit (Braun et al. 2019). CALIPSO is





included in this study to gain an insight into the reliability of its cloud climatology, in a context of sparse sampling and infrequent revisits.

— The EarthCARE mission is currently (January 2022) expected to launch in 2023. This CALIPSO follow-on mission will, like CALIPSO, use an afternoon orbit (equatorial crossing time 14:00 LST, ascending node), but follow a less-frequent revisit schedule ($n$=25 days). EarthCARE's along track lidar is expected to point 3° off-nadir, and operate

in the ultraviolet domain at 355 nm (Illingworth et al. 2015). EarthCARE is included in the present study to provide a general insight into CALIPSO data integrity, in terms of the revisit cycle.

— The Aeolus mission launched in 2018, and is designed to measure wind profiles with an ultraviolet (355 nm) Doppler lidar. The satellite is in an equatorial orbit with a crossing time at 18:00 LST (ascending node; dawn–dusk orbit). To meet mission requirements, Aeolus' across-track laser points 35° off-nadir, meaning that a measurement

track is located 230 km parallel to the ground track (Stoffelen et al. 2005). The present study, however, approximates that the lidar operates in nadir geometry. Aeolus is included for reference, as an example of a mission with a frequent revisit schedule ($n$=7 days).

— The ICESat-2 mission launched in 2018; the goal is to provide accurate laser altimetry for cryosphere and vegetation. The mission is characterized by a long revisit cycle ($n$=91 days), resulting in a dense ground track: path

separation at the equator is only 29 km. ICESat-2's altimeter operates in the visible domain (532 nm), its six laser beams are grouped in three pairs, separated by ~3.3 km. The actual pointing scheme changes between the polar regions and lower latitudes (see Markus et al. 2017 for details). In the present study, the configuration is simplified to one beam coinciding with nadir. ICESat-2 is included for reference, as an example of a mission with a very infrequent revisit regime.

For each mission, the simulation procedure was as follows.

1.    The ground track was generated based on orbital parameters, accessed in the form of Two-Line Elements (TLE). The ground track covered the full revisit cycle (of $n$ days), and provided geographical coordinates of sub-satellite points every one second. TLE for CALIPSO, Aeolus, and ICESat-2 were obtained from an online archive (celestrak.com). Elements for EarthCARE (yet to be launched) were kindly provided by Rob Koopman

and Montserrat Pinol Solé of the European Space Agency. Although EarthCARE TLE are fully representative of the final configuration, there is still one degree of freedom that will be defined post-launch (namely, the final choice of the longitude of the ascending node crossing at the equator), but this had no impact on the results of the present study.

2.    Each ascending and descending fragment of each orbit in a revisit cycle was individually projected onto the

Meteosat native coordinate system: namely, a vertical perspective from a geostationary orbit over 0° E.

3.    Every pixel in the Meteosat-projected transect was assigned cloud amount (CA), and cloud top height (CTH) data from the collocated Meteosat product. The assignment always used the Meteosat observation that was closest in time to the lidar's pass, taking the lidar's UTC time of the ascending/ descending node as reference.



The procedure was repeated for every day in a year; consequently, each ascending/ descending transect was
characterized by 365 (366) CA/ CTH values annually. A total of 10 years of Meteosat data were used for the
simulation (2007–2016), corresponding to 40 full cycles of ICESat-2 data, 146 of EarthCARE, 228 of
CALIPSO, and 521 of Aeolus.

4.  Simulated transects were then gridded into regular latitude-longitude grids of 1°, 2.5°, 5°, and 10°. These
    various resolutions were considered in order to evaluate how differences between $n$-day and 1-day climatology
relate to the spatial aggregation scheme.

5.  Finally, gridded transects were averaged into monthly and annual CA/ CTH values. At this stage, data were
    filtered in two ways: (1) the selection of transects that reflected the $n$-day revisit scheme; and (2) the selection
    of all data for all days (i.e. simulating a daily revisit).

The resulting (simulated) cloud climatology consisted of 120 monthly means, and 10 annual means, for two parameters (CA,
CTH), four satellites (CALIPSO, EarthCARE, Aeolus, ICESat-2), four grid resolutions, and two sampling scenarios (1-day,
$n$-day). It is important to note that no actual cloud data from the lidar missions were used. The only real information
exploited was the mission-specific ground track layout (orbital parameters). Actual CA/ CTH observations were replaced by
Meteosat data, in order to simulate a daily revisit. Since all lidar data were simulated, and all data originated from Meteosat,
the only differences between missions were due to different orbital configurations: namely, ground track density, revisit
frequency, and equatorial crossing time.

Statistical analyses focused on differences between the $n$-day and the 1-day climatology (mean difference, and mean absolute
difference). One-day means were always subtracted from $n$-day values: a positive difference indicates that $n$-day
observations overestimated the CA/ CTH value, whereas a negative difference represents the opposite case. If our hypothesis
is valid, differences should be close to, or equal to zero. Whenever the terms 'accuracy', or 'error' are used, they refer to
Meteosat 1-day estimates, which are assumed to be ground truth for the purposes of this study

## 3 Results

### 3.1 Differences between a 1-day and an $n$-day revisit regime

The analysis found that mean global differences between 1-day (simulated-hypothetical) and $n$-day (simulated-actual) cloud
climatologies were insignificant. Specifically, they never exceeded 0.1 % for CA, and 10 m for CTH – regardless of the
satellite mission, spatial resolution of the final grid, or timeframe of averaged data (monthly or annual).

However, the data aggregation approach did find differences between 1-day and $n$-day datasets when they were interpreted
in terms of absolute values. Here, the magnitude of discrepancies was largest at the finest resolution, and decreased as the
grid box size increased. Regarding monthly means (Tab. 2), there was a decrease from ~11 % at 1° to ~2 % at 10° for CA,
and from ~1500 m at 1° to ~300 m at 10° for CTH. The same trend was observed for annual means, except that the absolute
difference was 3–4 times lower compared to monthly values.





**Table 2. Difference in mean global cloud amount (CA), and cloud top height (CTH) between a hypothetical 1-day revisit mission, and the actual n-day revisit. Single-year monthly and annual means for 2005–2016 are considered**

| Lidar mission | Differences in monthly means | | | | | | Differences in annual means | | | | | |
| | CA (%) | | | CTH (m) | | | CA (%) | | | CTH (m) | | |
| | diff | std | \|diff\| | diff | std | \|diff\| | diff | std | \|diff\| | diff | std | \|diff\| |
| --- | --- | --- | --- | --- | --- | --- | --- | --- | --- | --- | --- | --- |
| *1×1° grid box size* | | | | | | | | | | | | |
| Aeolus | 0.01 | 11.2 | 8.6 | 4 | 1544 | 1150 | 0.02 | 3.2 | 2.5 | −1 | 435 | 331 |
| CALIPSO | 0.04 | 15.3 | 11.6 | 6 | 2071 | 1562 | 0.04 | 4.3 | 3.3 | 6 | 610 | 458 |
| EarthCARE | −0.02 | 15.0 | 11.5 | −3 | 2091 | 1582 | −0.01 | 4.2 | 3.3 | 1 | 605 | 460 |
| ICESat | 0.01 | 16.0 | 12.3 | -10 | 2212 | 1686 | 0.02 | 4.4 | 3.5 | −1 | 643 | 490 |
| *2.5×2.5° grid box size* | | | | | | | | | | | | |
| Aeolus | 0.01 | 7.7 | 5.9 | −2 | 1092 | 810 | 0.01 | 2.2 | 1.7 | −2 | 302 | 232 |
| CALIPSO | 0.03 | 7.8 | 6.0 | 7 | 1145 | 860 | 0.03 | 2.2 | 1.7 | 5 | 315 | 245 |
| EarthCARE | −0.02 | 7.7 | 5.9 | 2 | 1131 | 848 | −0.01 | 2.2 | 1.7 | <1 | 313 | 242 |
| ICESat | 0.02 | 8.4 | 6.5 | -3 | 1257 | 935 | 0.02 | 2.4 | 1.9 | −3 | 347 | 267 |
| *5×5° grid box size* | | | | | | | | | | | | |
| Aeolus | 0.01 | 4.8 | 3.7 | −4 | 676 | 506 | 0.01 | 1.4 | 1.0 | −3 | 188 | 145 |
| CALIPSO | 0.04 | 4.7 | 3.6 | 6 | 688 | 520 | 0.04 | 1.3 | 1.1 | 5 | 191 | 149 |
| EarthCARE | <0.01 | 4.6 | 3.5 | 1 | 674 | 509 | <0.01 | 1.3 | 1.0 | <1 | 187 | 146 |
| ICESat | 0.03 | 4.6 | 3.6 | <1 | 716 | 534 | 0.03 | 1.3 | 1.0 | −3 | 169 | 152 |
| *10×10° grid box size* | | | | | | | | | | | | |
| Aeolus | 0.02 | 2.7 | 2.1 | −6 | 370 | 281 | 0.02 | 0.7 | 0.6 | −5 | 105 | 82 |
| CALIPSO | 0.04 | 2.6 | 2.0 | 4 | 381 | 292 | 0.04 | 0.7 | 0.6 | 4 | 106 | 83 |
| EarthCARE | −0.02 | 2.5 | 1.9 | −1 | 360 | 279 | −0.02 | 0.7 | 0.5 | −1 | 99 | 77 |
| ICESat | 0.03 | 2.8 | 2.1 | <1 | 429 | 325 | 0.03 | 0.7 | 0.6 | −2 | 117 | 91 |

At 1° resolution, Aeolus performed slightly better than the other missions. In this case, monthly mean *n*-day and 1-day
estimates differed (on average) by 8.6 % (CA), and 1150 m (CTH). Corresponding discrepancies for other the lidar missions
were in the range 11–12 %, and 1562–1686 m. However, as the resolution fell to 2.5° (or lower), discrepancies noted for
Aeolus reached levels observed for CALIPSO and EarthCARE. On the other hand, ICESat observations appeared to be least
accurate – but only at 1° and 2.5° – where the mission reported slightly higher discrepancies than EarthCARE or CALIPSO.
At coarser resolutions, statistics were in close agreement for all four missions for 1-day and *n*-day climatologies.

A more detailed overview of the actual agreement between mission scenarios was revealed by the distributions of
differences. Negligibly small differences in CA and CTH for global means (Tab. 2) suggested no bias in the data, and Fig. 1
highlights that they were only an effect of error balancing. Infrequent sampling (every *n* days) led to a CA overestimation of
2–4 %, and a CTH underestimation of 100–200 m, according to the location of the peak frequency in distributions.

The highest monthly differences exceeded ±30 % (CA), and ±2 km (CTH) at 1° resolution. As expected, the spread of the
differences (shape of the distribution) was a function of the observation aggregation strategy. Distributions were asymmetric

at 1° resolution, and became more Gaussian for larger grid box sizes. The normalisation of distributions was especially evident for monthly means (Fig. 1 a–f), while annual means were always symmetric and close to normal (Fig. 1 g–l).

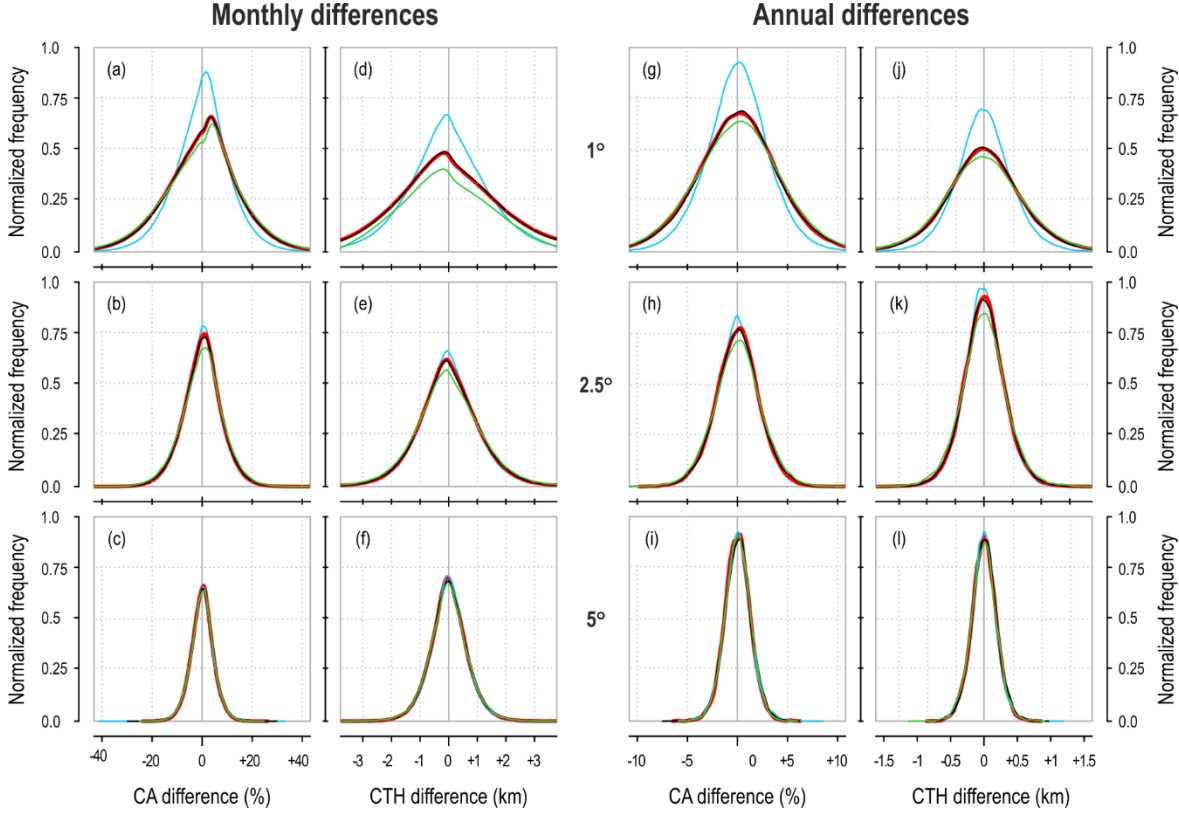

**Figure 1. Differences between n-day and 1-day revisit cloud amounts, and cloud top heights at monthly (left panel), and annual (right panel) mean timescales, for CALIPSO (red), EarthCARE (black), Aeolus (blue), and ICESat-2 (green). In all cases, n-day values are subtracted from 1-day values.**

A decrease in spatial resolution also resulted in the convergence of distributions. At 5°, the frequency of differences in CA/ CTH was almost identical (Fig. 1 c, f, i, l), both for monthly and annual means. At such a high level of spatial generalisation, the actual revisit time of a profiling lidar ($n$=7, 16, 25, or 91 days) had no significant impact on CA/ CTH estimations.

The distribution of differences in CA/ CTH for annual means confirmed the (dis)similarities between missions already noted for globally-averaged data. Interestingly, two missions (namely CALIPSO and EarthCARE) shared nearly exactly the same distribution. Despite their different revisit frequencies (16 and 25 days), both missions reproduced the 1-day cloud climatology with the same accuracy, both for monthly and annual timescales, and regardless of the spatial resolution of the target global grid.

Geographical regions where discrepancies in CA/ CTH between $n$-day and 1-day climatologies were smallest are highlighted in Fig. 2. Regarding CA, the best agreement was noted for oceans at high latitudes (>45° N/S), where the mean absolute




**Figure 2. The geographical distribution of mean absolute differences between *n*-day and 1-day revisit cloud amounts, and cloud top heights at monthly mean (left panel), and annual mean (right panel) timescales. In all cases, *n*-day values are subtracted from 1-day values.**




difference between datasets typically did not exceed ~5 % on the monthly time scale. Lower latitudes featured absolute differences ranging between 5 % and 10 %, with only local exceptions of 10–15 % (e.g. Spain, north Argentina, small parts of the east coast of Africa). Absolute errors in CTH estimations were largest over north Africa (the eastern Sahara Desert), exceeding ~1.5 km. On the other hand, the best agreement in CTH was found for the eastern Atlantic, at 0–30° N (a region

of frequent marine stratocumulus), where CTH were reported to ±0.5 km accuracy, or better.

The geographical distribution of absolute differences in CA/ CTH estimations was controlled by two major relationships. First, the magnitude of differences depended on average CA for a location. In very cloudy, or almost cloudless regions, discrepancies in CA estimations were smallest (Fig. 3a), since their presence could be probed with the same efficiency, regardless of the revisit schedule. On the other hand, differences in CTH tended to decrease as CA increased, over the full

range of cloudiness (Fig. 3e). The second relationship linked the discrepancy in a parameter estimation with the number of observations. As expected, the more frequently a grid box was sampled by a lidar, the lower the absolute difference between *n*-day and 1-day climatologies both for CA (Fig. 3c), and CTH (Fig. 3g).

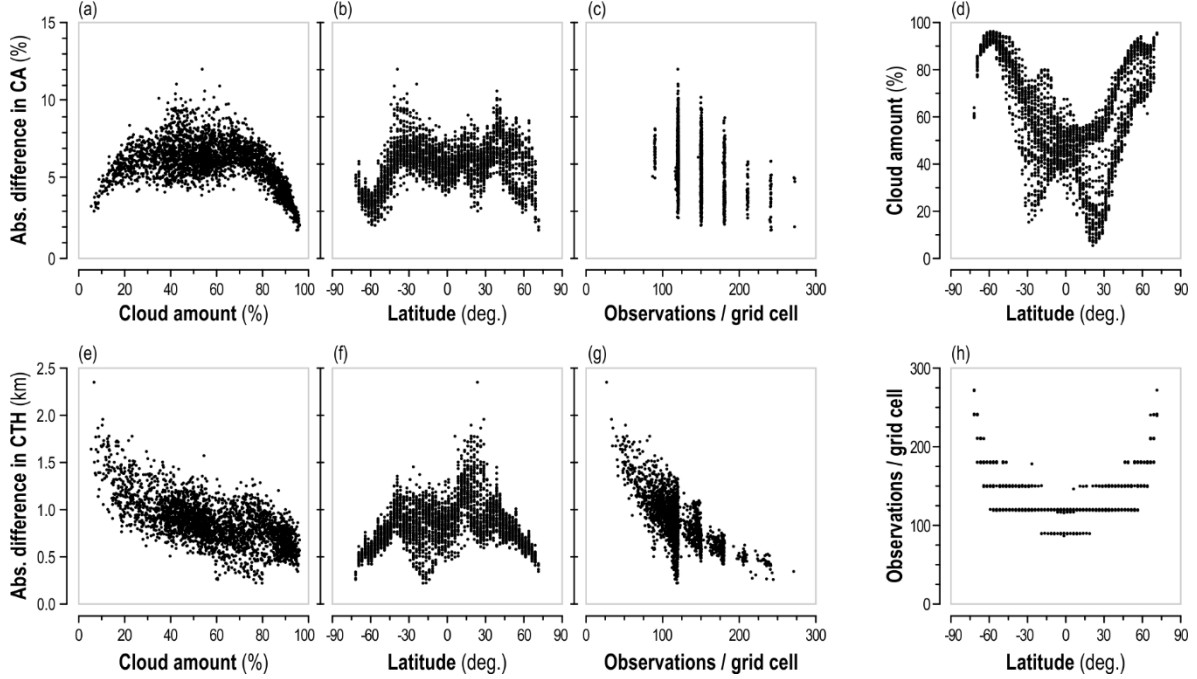

**Figure 3. The relation between the mean annual difference in cloud amount (a–c)/ cloud top height (e–g) between n-day and 1-day revisit climatologies, and: mean cloud amount (a/ e), latitude (b/ f), and the number of observations in the 1-day revisit scenario (c/ g). For reference, cloud amount in relation to latitude (d), and the number of observations (h) is shown. Data refer to the simulated 2.5° resolution CALIPSO dataset.**





Since both average CA (Fig. 3d), and the number of observations (Fig. 3h) depended on latitude, the related parameters (CA, CTH) showed a similar dependency (Fig. 3 b, f) modulated at the region level by CA (and the underlying global-scale circulation patterns). Specifically, the 1-day climatology was most accurately reproduced by *n*-day data at high latitudes (high CA and frequent revisits are consistent with smaller differences), but with lower accuracy at lower latitudes (higher differences are due to less frequent revisits, regionally modulated by variable CA). This spatial pattern was observed across

all investigated missions, both for monthly (Fig. 2, left panel), and annual means (Fig. 2, right panel). This variance in the magnitude of differences between missions, resolutions and timescales reflects trends observed for cloud distribution (Fig. 1).

**3.2 Exploring an alternative revisit frequency for a CALIPSO-like mission**

The results presented so far relate to satellite missions with a specific orbital configuration: namely, a fixed revisit frequency

(every *n* days), and a native ground track layout. For instance, in the case of CALIPSO, differences in CA and CTH were calculated for 1-day and 16-day climatologies, evaluated as spatial and temporal aggregations of instantaneous observations. However, another interesting question is how the selection of an *n* value itself impacts differences in CA/CTH. This question was answered with a test focused on the CALIPSO mission alone.

Simulation runs for CALIPSO assuming *n*=1 (a reference, high temporal resolution climatology) and *n*=16 (the actual revisit

frequency) were supplemented with an additional 14 runs for all *n* between 1 and 16. Next, CA and CTH values for the 1-day revisit were subtracted from each *n*-day revisit estimate. The resulting statistics are summarized in Fig. 4. This figure shows that absolute differences in both CA and CTH increased as the revisit period increased. Importantly, the change was gradual, with no rapid variation in discrepancies. Regarding the impact of spatial resolution, a general rule was that doubling the grid box size reduced the range of differences by half for all *n*.

At the finest spatial resolution (1°) the spread of differences in CA was so large that even a 2-day revisit would not be enough to keep them within a ±1 % range. However, CALIPSO observations are most frequently gridded at 2.5° resolution. In this case, a constellation of six CALIPSO-like observatories at adequately phased orbits would be sufficient to provide ±1 % accuracy for ~80 % of locations (grid boxes) in the study area. It should be noted, however, that a 2-day revisit (equivalent to eight satellites) would be necessary to achieve 1 % accuracy in CA for all locations. Similar tendencies were observed for

CTH at 2.5°. To obtain CTH statistics that differ from a 1-day climatology by no more than ±150 m, a constellation of eight CALIPSO-like missions would be required (equivalent to a 2-day revisit). A 3-day revisit schedule resulted in ±150 m accuracy for 80 % of locations in the study area. At coarser resolutions (5–10°) there was no substantial decrease in CTH accuracy (with respect to a 1-day revisit) when *n* was six days or more. On the other hand, it should be noted that CALIPSO data gridded at ≤5° are only used occasionally.


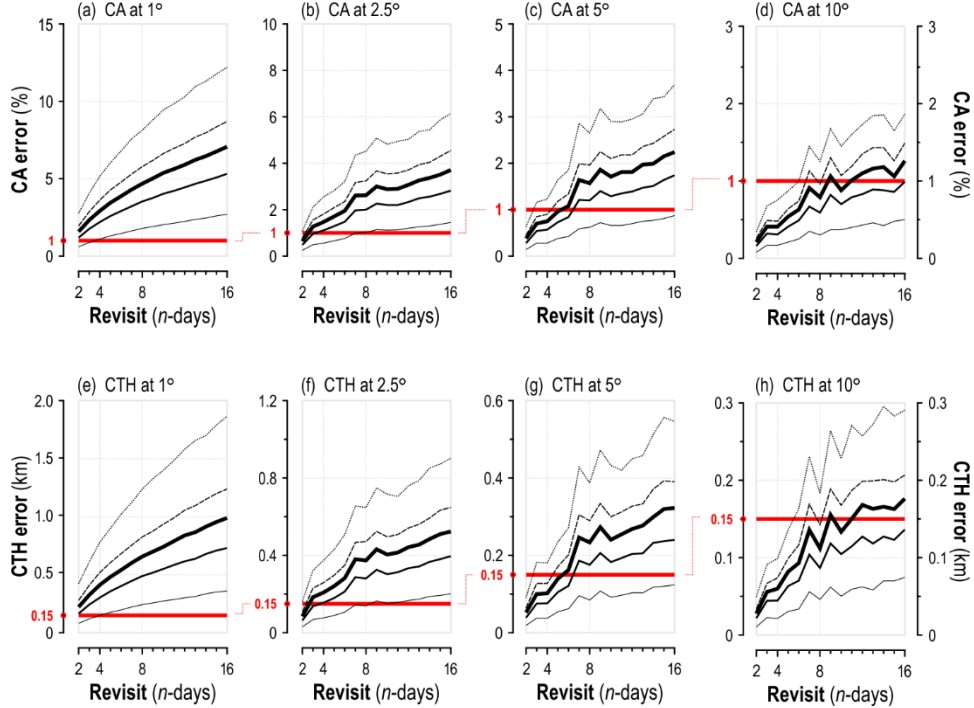

**Figure 4. The spread of differences in cloud amount (top panel) and cloud top height (bottom panel) between 1-day and n-day revisit climatologies, with respect to n ranging from 1–16 days. Data refer to the simulated 2.5° resolution CALIPSO dataset. Starting from the top-most, the lines indicate range covering: 99% of observations (dotted line), 95% (dashed line), 90% (bold solid line), 80% (regular solid line), and 50% (thin solid line). Red bar indicated location of the accuracy threshold (see text for details). 16-day revisit can be achieved with one CALPSO-like satellite, 8-day with two, 4-day with four, and 2 day with eight (phased orbits assumed). Note that for clarity the vertical scale has been adjusted for each plot.**

### 3.3 Offset of the initial day of a revisit cycle

The revisit cycle of any mission in a sun-synchronous orbit is 'anchored' to the satellite's launch date. For instance, CALIPSO started its 233-orbit cycle on April 28, 2006 – the day the satellite was deployed. Consequently, the annual mean of a cloud property reported by CALIPSO relies on data observed at a specific time and location, defined by a ground track that was initiated on that day (and repeated every 16 days ever since).

When deriving a cloud climatology, it is assumed that the temporal offset related to the start day of a revisit cycle should have no impact on the annual mean of a cloud parameter. That may be true for wide-swath imagers, but is questionable for profiling lidars, when the annual mean is based on as few as 22–23 observations captured every 16 days (in the case of CALIPSO). In order to validate the latter assumption in the context of a lidar mission, the following experiment was performed.

For each year, mean annual CA and CTH were calculated using the CALIPSO ground track for its 16-day revisit schedule. First, statistics were calculated for a revisit pattern that agreed with real-time mission overpasses (using the actual launch



date of the satellite). Next, the same procedure was repeated 15 times, while in each iteration the pass date was incremented by one to replicate the situation where CALIPSO had been launched 1, 2, ..., 15 days later than it actually was. Therefore, each location was characterized by 16 CA/CTH estimates, covering all possible launch dates. Finally, the highest and lowest values were examined. If the tested assumption was true, the difference between maximum and minimum values should be

close to zero.

**Table 3. Cloud amount (%) for simulated CALIPSO-like scenarios, calculated for a single year and different locations (top rows), and multiple years at a single location (bottom rows). See text for details.**

| Grid box location / year | actual day | 16-day revisit range for all possible days | location within range (%) | 1-day revisit |
|---|---|---|---|---|
| *Single year, many locations* | | | | |
| 16.5° N, 22.5° E / 2006 | 9.6 | 9.6-19.2 | 0 | 14.6 |
| 63.5° N, 20.5° E / 2006 | 76.3 | 58.1-76.3 | 100.0 | 68.5 |
| 35.1° N, 53.5° E / 2006 | 87.6 | 82.4-97.3 | 53.5 | 88.7 |
| 41.5° N, 31.5° E / 2006 | 54.8 | 41.4-65.5 | 55.5 | 55.0 |
| *Single locations, various years* | | | | |
| 41.5° N, 31.5° E / 2006 | 54.8 | 41.4-65.5 | 55.5 | 55.0 |
| 41.5° N, 31.5° E / 2007 | 42.8 | 42.8-56.8 | 0.0 | 49.3 |
| 41.5° N, 31.5° E / 2008 | 43.5 | 37.2-54.4 | 36.8 | 47.5 |
| 41.5° N, 31.5° E / 2009 | 53.7 | 41.4-64.9 | 52.4 | 51.1 |
| 41.5° N, 31.5° E / 2010 | 54.1 | 47.9-62.0 | 43.8 | 55.8 |
| 41.5° N, 31.5° E / 2011 | 59.8 | 37.2-60.5 | 97.1 | 52.6 |
| 41.5° N, 31.5° E / 2012 | 60.8 | 38.6-60.8 | 100.0 | 49.9 |
| 41.5° N, 31.5° E / 2013 | 42.7 | 39.0-67.4 | 13.2 | 51.7 |
| 41.5° N, 31.5° E / 2014 | 61.2 | 46.4-68.3 | 67.8 | 55.0 |
| 41.5° N, 31.5° E / 2015 | 44.8 | 34.7-68.0 | 30.3 | 53.2 |

Table 3 shows the results of the simulation at sample locations. For instance, a 1° grid box centred at 16.5° N, 22.5° E reported maximum CA of 19.2 %, and a minimum of 9.6 %. Since CA for the actual pass day is known (9.6 %) it can be concluded that the mission coincided with the lowest of all 16 estimates. If the CALIPSO mission had been launched a few days later, the reported value would have been 19.2 %. It is important to note that although both values (as for any min-max range) are equally valid, only one was reported in the CALIPSO climatology – and then used in numerous applications.

For a grid box centred at 41.5° N, 31.5° E the spread of possible CA values in 2006 ranged from 41.4 % to 65.5 %, and the annual mean reported for the actual pass date (54.8 %) was in the middle of the range. However, in subsequent years, the value for the actual pass date matched the highest (2012), or the lowest (2007) estimate out of the 16 possible scenarios. It is only when CALIPSO transects are averaged into a 10-year annual mean that the value for the actual pass date will be in the middle of the min-max range, essentially being the value estimated for a 1-day revisit.

The results of the simulation for all locations in the study area are mapped in Fig. 5. Annual geographical distributions of differences between maximum and minimum CA/ CTH duplicate the pattern reported for discrepancies between 1-day and





*n*-day climatologies (Fig. 5 c, d). Similarly, the width of the min-max range is a function of the spatial resolution of the target grid. For CA, it was typically over 14 % at 1°, and gradually narrowed to 7 % at 2.5°, and 3 % at 10° resolution (Fig. 5a). The corresponding change in CTH was from 1920 m (1°) to 1030 m (2.5°), and 370 m (at 10°) (Fig. 5b).


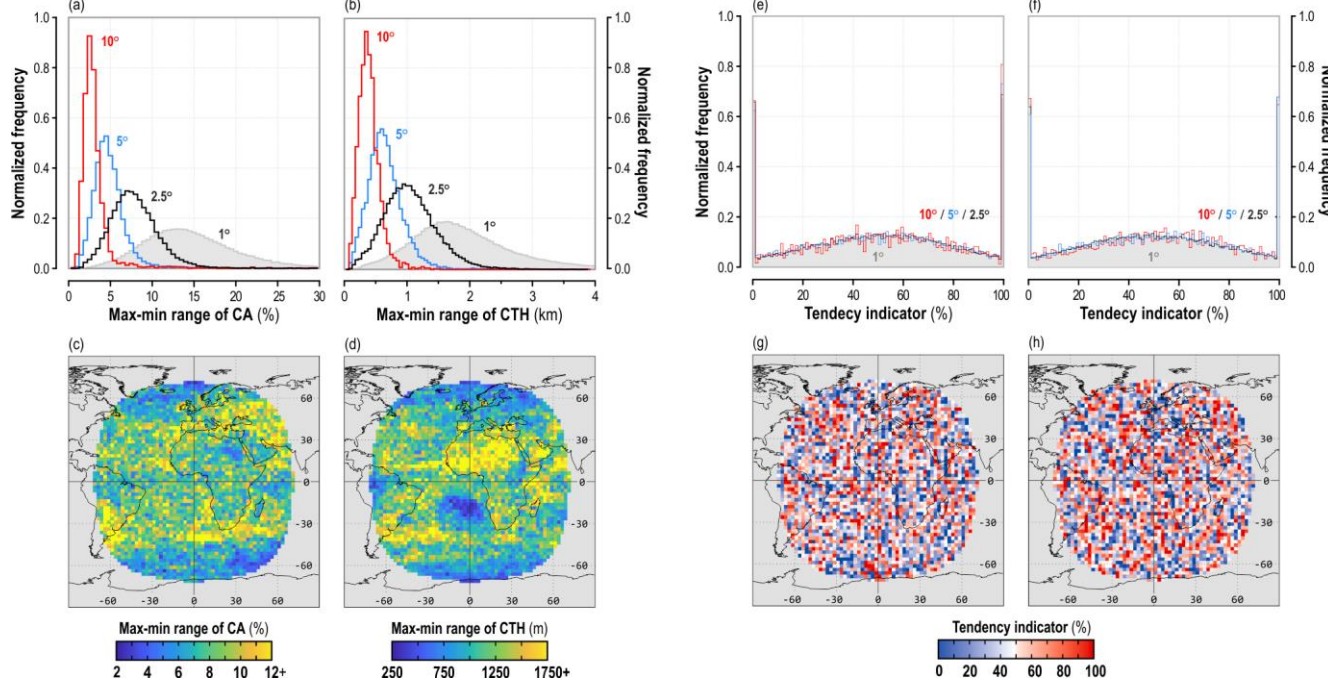

**Figure 5.** The range between maximum and minimum estimates of mean annual cloud amount (c), and mean annual cloud top height (d) for all of the possible start dates for the CALIPSO ground track. The tendency for a parameter in the CALIPSO dataset from the actual launch date to be over/ underestimated is shown in (g) for cloud amount, and (g) for cloud top height. Maps refer to the simulated 2.5° resolution CALIPSO dataset for 2008


A dedicated indicator was introduced to map the coincidence between CA/ CTH values on actual CALIPSO pass dates, and the max/min value of a parameter for all 16 investigated scenarios. The min-max range was scaled to 0–100 %, and the CA/ CTH value for the actual pass date was expressed on that scale. For example, 0 % meant that an estimate matched the lowest possible value, 100 % indicated that it matched the highest possible value, and at 50 % it agreed with the middle of the range.


The geographical distribution of the indicator was random for both CA (Fig. 5 g) and CTH (Fig. 5 h), thus two adjacent grid boxes could represent dramatically different tendencies. For a single year, it was unlikely that the annual mean would coincide with the middle of min-max range: only a quarter of locations reported a value between 40 % and 60 % (Fig. 5 e, f). For up to ~12 % of locations the CA/ CTH value for the actual CALIPSO pass date represented the most extreme case (maximum and minimum CA/ CTH values were matched in 16 scenarios). The randomness of the geographical distribution of the indicator, along with the statistical distribution of its value, had no relation to the spatial resolution of a grid.






## 4 Discussion

If we assume that a daily revisit regime represents true mean CA/ CTH, the results of this study allow us to investigate the
accuracy of lidar-based cloud climatologies used in climate change applications. Ohring et al. (2005) set two goals for
satellite instrument calibration: 1 % for total CA, and 150 m for CTH. The latter values correspond well with upper accuracy
thresholds set by the Global Climate Observing System for satellite-based data products (WMO 2011).

**Table 4. The fraction (%) of locations (specifically, grid boxes within the Meteosat field of view) that met the accuracy criterion of**
**1% for cloud amount (CA), and 150 m for cloud top height (CTH), at monthly and annual timescales.**

| Lidar mission | Monthly | | Annual | |
|---|---|---|---|---|
| | CA | CTH | CA | CTH |
| *1×1° grid box size* | | | | |
| Aeolus | 9 | 10 | 27 | 31 |
| CALIPSO | 6 | 7 | 20 | 23 |
| EarthCARE | 6 | 7 | 20 | 22 |
| ICESat | 6 | 6 | 19 | 21 |
| *2.5×2.5° grid box size* | | | | |
| Aeolus | 12 | 14 | 39 | 42 |
| CALIPSO | 12 | 13 | 37 | 40 |
| EarthCARE | 12 | 13 | 37 | 40 |
| ICESat | 11 | 12 | 35 | 37 |
| *5×5° grid box size* | | | | |
| Aeolus | 19 | 21 | 58 | 61 |
| CALIPSO | 19 | 20 | 56 | 59 |
| EarthCARE | 20 | 21 | 59 | 60 |
| ICESat | 19 | 20 | 57 | 59 |
| *10×10° grid box size* | | | | |
| Aeolus | 33 | 36 | 84 | 85 |
| CALIPSO | 32 | 34 | 83 | 84 |
| EarthCARE | 34 | 35 | 86 | 87 |
| ICESat | 31 | 31 | 82 | 82 |

Our study revealed that 1 % (CA) and 150 m (CTH) thresholds are beyond the capability of all investigated lidar missions,
either at monthly or annual timescales (Tab. 4). As few as ~5–15 % of locations (grid boxes) in the Meteosat domain met the
1 % criterion for CA and 150 m for CTH at the monthly scale, and at 1–2.5° spatial resolution. When gridded at a coarser
resolution of 10°, roughly one third of grid boxes met CA/ CTH accuracy criteria. Only when annual mean CA/ CTH was
considered did the majority (~60 %) of grid boxes meet the requirements at (low) 5° spatial resolution.

Importantly, our study (see Tab. 2) only considered single-year monthly and annual means. On the other hand, the CALIPSO
mission has provided ~15 years of data. Table 4 suggests that when up to 10 years of observations are aggregated, it is
realistic to expect 1 % accuracy in mean annual CA. However, this does not apply to the 10-year monthly mean: here, the



total number of observations is comparable to the single-year annual mean, and therefore the magnitude of uncertainty will be similar to this mean.

One possible method to reduce errors is to increase the number of observations per year/ month by increasing the number of satellites. However, as our CALIPSO-oriented experiment showed, the strategy will not be efficient in most cases. A 1 % (CA) or 150 m (CTH) accuracy for all 1° or 2.5° grid boxes would still require a daily revisit, meaning as many as 16

CALIPSO-like missions. Such a constellation is unlikely given the current CALIPSO technology and costs. However, the ongoing revolution in satellite technology (Stephens et al. 2020) and data processing (Yorks et al. 2021) may result in cost-effective small missions in the near future (e.g. the Time-varying Optical Measurements of Clouds and Aerosol Transport satellite, or the Methane Remote Sensing Lidar Mission; Ehret et al. 2017).

The second important finding reported here is that all lidar missions differed from the 1 day climatology in almost the same

way (Fig. 1, Fig. 2), despite their revisit frequency – however, only when the spatial resolution of the target grid was close to 5° or less. This means that at lower spatial resolution (larger grid box), an infrequent revisit schedule is compensated for by a denser ground track, and the effective number of observations is similar for all missions.

Our findings do not imply that CA and CTH from all missions will be exactly the same in real life. Lidar sensors onboard satellites differ in terms of both construction and their cloud detection algorithms. Furthermore, different satellites operate in

orbits with different equatorial crossing times, therefore, CA/ CTH values refer to different moments of the diurnal cycle (Noel et al. 2018).

In this context, CALIPSO and EarthCARE may be important exceptions. When launched, EarthCARE will be placed in an orbit with an equatorial crossing time at 14:00 local solar time, only 30 minutes after CALIPSO. Although the two missions will differ in terms of their equipment, the scientific community is seeking to standardize cloud detection algorithms, and

make the final products coherent (Okamoto and Sato 2018). If these efforts are successful, the only significant difference between the two missions will be their revisit frequency: 16 days for CALIPSO, and 25 days for EarthCARE. As this study has shown, this factor alone is not sufficient to result in significant differences. On the contrary, the two missions should provide almost the same CA/ CTH statistics, and the EarthCARE mission's cloud climatology should be consistent with CALIPSO.

Finally, our study – for the very first time – has evaluated the magnitude of possible uncertainties resulting from the infrequent sampling regime of lidar missions, based on realistic cloud data (empirical cloud regimes, and how they are distributed geographically). Our results suggest that some practical considerations should be taken into account when using lidar data to validate other cloud climatologies or models.

According to Winker et al. (2017), nadir-only observations should (theoretically) provide sufficient sampling accuracy at

monthly and annual global scales. Specifically, root mean square error for CA should be <1 % (Ohring et al. 2005). The present study confirmed that this thesis is valid for all investigated lidars, and also demonstrated that 1 % accuracy can be achieved for ~80 % of locations at 10×10°, assuming 1-day revisit data as true CA (Tab. 4). However, at the finer resolutions





typically used to investigate the geographical distribution of CA/ CTH (1–2.5°), accuracies were much lower, and failed to meet required standards.

Our CALIPSO-focused experiments (Sect. 3.3) demonstrated that the single-year annual mean (at grid box level) is very sensitive even to theoretically irrelevant aspects such as the initial day of the revisit cycle. Shifting this date backward or forward by one day may result in a significantly different estimate of mean annual CA, and this finding must be taken into account when CALIPSO data are used for validation. For instance, Heidinger et al. (2012) compared 2007 mean annual CA from the Advanced Very High Resolution Radiometer to CALIPSO estimates for the same year. Locally, differences

between datasets were up to 10–20 %. However, the present study revealed that as much as 15 % of the difference in CA can be attributed to uncertainty related to CALIPSO's infrequent and sparse sampling regime. Similarly, Franklin et al. (2013) used mean seasonal CA from CALIPSO as validation data, but neglected uncertainty in lidar estimates at such a short timescale. Finally, Chepfer et al. (2008) compared day and night CA from CALIPSO using 3-month means. Although discrepancies reached ±25 %, the authors did not discuss the impact of infrequent sampling.

Sparse sampling and an infrequent revisit schedule have most impact on short-term, lidar-based cloud climatologies. Therefore, whenever a single-year annual mean is validated, the resulting lidar climatology should not be considered as a point estimate (mean value), but rather as a confidence interval for the mean (e.g. Kotarba and Solecki 2021). An alternative approach is to match individual lidar profiles with imagers on a per-pixel basis (Wang et al. 2016; Kotarba 2020; Heidinger et al. 2012). Unfortunately, this is only possible in the rare cases where the lidar's and imager's orbits are aligned.

More research is needed to test how the lidar's revisit schedule should be accounted for in satellite simulators (e.g. the CFMIP Observation Simulator Package, COSP; Bodas-Salcedo et al. 2011). The goal of simulators is to reduce bias between satellite-observed and model-generated cloud parameters. This is achieved by producing satellite-like radiances from a modelled atmosphere, then retrieving sensor-specific data for the desired geophysical variable. Unfortunately, satellite sampling factors (the revisit frequency, the ground track density) are typically neglected in satellite simulators.

Consequently, simulated lidar-like cloud parameters resemble a daily revisit mission, rather than actual *n*-day sampling. This source of bias remains largely unaddressed. On the other hand, model validations tend to use long time series (10+ years) of CALIPSO observations, which greatly increases the number of observations that shape the annual mean. As a consequence, revisit-related uncertainties become relatively limited.

## 5 Summary and Conclusion

This study is the first of its kind to compare lidar cloud climatologies (CA, CTH) for a hypothetical 1-day revisit, and the actual *n*-day revisit regime (where *n* is mission specific). We considered four missions: CALIPSO (n=16), EarthCARE (*n*=25), Aeolus (*n*=7), and ICESat-2 (*n*=91), with a special focus on CALIPSO, as it has contributed most to cloud climatology studies. Statistics for 10 years of observations (2007–2016) were evaluated, taking into account both spatial (ground track density), and temporal (the actual temporal sequence of the orbit during the revisit cycle) sampling scenarios.





Highly-reliable, Meteosat 15-minute data were used as the source of real CA and CTH, and to understand how the two parameters actually vary in time and space (realistic cloud regimes).

Our central hypothesis was that cloud parameters estimated from an *n*-day revisit mission do not differ significantly from values that would be obtained with a daily revisit schedule. The results of our simulation demonstrate that this hypothesis is invalid for most of the evaluated circumstances. Specifically:

— assuming a 1-day revisit regime as a proxy for true CA/ CTH, the actual (*n*-day) revisit schedule was insufficient to calculate mean annual values of parameters that met required accuracies (1 % for CA, 150 m for CTH), at a spatial resolution above 10° latitude and longitude. This required accuracy was only achieved at the mean global scale, and only for most (~80 %) 10×10° grid boxes;

— mean annual CA is very sensitive to the revisit frequency, and the corresponding ground track density. For a single
year, revisit-related uncertainties for CA/ CTH can be as high as 15 %/ 1800 m (on average) when lidar transects are gridded at 1° resolution, or 5 %/ 1000 m when gridded at 2.5° resolution. As a consequence, whenever lidar data are used to validate other cloud datasets (either empirical or modelled), the revisit time should be accounted for by using, for example, confidence intervals instead of point estimates (mean, median);

— despite their different revisit frequencies, we found a similar magnitude of discrepancy for all of the investigated
lidars (CALIPSO, Aeolus, ICESat-2, EarthCARE), between the hypothetical 1-day revisit mission, and the actual *n*-day revisit climatology – especially at lower resolutions (≤2.5°). The latter finding suggests that at this scale of aggregation, infrequent revisits are compensated for by the ground track density. It also suggests that CALIPSO, together with its follow-on mission EarthCARE, are very likely to produce consistent cloud records despite the difference in sampling frequency.

The present study implemented a simulation method that can test uncertainties in individual lidar missions, or tandem polar-orbiting + inclined-orbit lidar constellations (CALIPSO–CATS, or the AOS-P1–AOS-I1 concept that has been studied under NASA's Atmosphere Observing System, http://aos.gsfc.nasa.gov/). When a single lidar mission is considered, the method can also be used globally with a polar-orbiting imager that shares the same revisit frequency and equatorial crossing time (currently this is only possible for CALIPSO, simulated with MODIS/Aqua). If this is not possible, any geostationary
platform or high-frequency atmospheric model can be used instead.

**Data availability**

Data analyzed in this study were a re-analysis of existing data. METEOSAT cloud data are openly available from the EUMETSAT Satellite Application Facility on Climate Monitoring (CM SAF; https://www.cmsaf.eu/).



**Author contribution**

AZK designed the research, carried it out, and prepared the manuscript.

**Competing interests**

The authors declare that they have no conflict of interest.

**Acknowledgements**

This study was funded by the National Science Centre of Poland grant no. UMO-2017/25/B/ST10/01787 and partially
supported by PL-Grid Infrastructure (computing resources).

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
