# Peer review of "Impact of the revisit frequency on cloud climatology for CALIPSO, EarthCARE, Aeolus, and ICESat-2 satellite lidar missions"

_Atmospheric Measurement Techniques, 2022_

## Referee Comment (RC2)

**A Review of "Impact of the revisit frequency on cloud climatology for CALIPSO, EarthCARE, Aeolus, and ICESat-2 satellite lidar missions" by Andrzej Kotarba**

This paper investigates the satellite sampling of cloud amount and cloud top height from several sun-synchronous satellites hosting lidar instruments compared to SEVERI. The paper is well written, clear, and provides results that are important for designing future space-based lidar mission architectures. It deserves to be published after a few minor revisions that I believe will strengthen the paper.

My 2 major comments are:

1) Diurnal variability: It is not clear to me what role diurnal variability plays in this sampling study. Based on #3 on line 156, I believe only Meteosat data from the time of the satellite overpass was used in the "truth" dataset. Is that true? If so, your monthly and annual averages are biased to the times of the overpasses and thus diurnal variability will not be accounted for. That is fine for assessing the sampling at the equatorial crossing times of each satellite, but that needs to be explicitly stated in the paper. If the SEVIRI "truth" dataset DOES include clouds from all times of day, then the geographical distribution of absolute differences in CA/ CTH estimations (Figure 2) will be influenced by diurnal variability of CA or CTH. This pattern (1-day climatology was most accurately reproduced by n-day data at high latitudes but with lower accuracy at lower latitudes) would also be more consistent with where the largest diurnal variability was reported by Noel et al. (2018).
2) Big picture impacts: The author does a great job discussing what these results mean for future lidar missions in the Conclusion. But after reading the paper, I found myself asking - what does this means for current data users? For example, if I want to use the data from these missions to compute global, annual cloud radiative effects based on CA and CTH, I can do that confidently. However, if I want to compute radiative effects at seasonal/monthly and regional/finer spatial scales, the CA and CTH from these lidar datasets may be biased based on Table 4. That is an important point for data users and will make this paper worth citing for future authors. I suggest adding a few sentences on this topic to the Discussion or Conclusion.

The 4 minor comments to be addressed are:

1) Table 1: Do all these satellites have a 98-degree inclination angle? If not, I suggest adding an inclination angle column to this table since it impacts the repeat times.
2) Line 66: I suggest citing Yorks et al 2016 for CATS as it is more of an overview paper. The citation is: Yorks, J. E., M. J. McGill, S.P. Palm, D. L. Hlavka, P.A. Selmer, E. Nowottnick, M. A. Vaughan, S. Rodier, and W. D. Hart (2016), An Overview of the CATS Level 1 Data Products and Processing Algorithms, Geophys. Res. Let., 43, doi:10.1002/2016GL068006.
3) Line 110-111: Are there any papers that reports the accuracy of CA and CTH from the SEVIRI products? If so, I suggest adding a sentence to report those accuracies and cite

those papers. I know that doesn't impact the results of the study, but I found myself wondering what the accuracies are as I read the paper.

4) Line 445-446: This is a highly relevant point for future architecture designs. Did you consider looking at the ISS to provide a reference point for a lower inclination angle? I know it would be more work, but I think it would really strengthen the paper to add the ISS to this analysis. At the very least, it would be beneficial to add a sentence or two about the ISS revisit time and where it may fall compared to the satellites you studied.

---

## Author Comment (AC1)

**RC1: 'Comment on amt-2022-80', David Winker, 26 Apr 2022**

This paper examines whether the impacts of sparse sampling from a nadir-viewing satellite lidar varies with the revisit time of the satellite orbit. Orbits of several different existing satellite lidars are chosen as examples. Parameters of interest are cloud amount, cloud top height, and cloud optical depth. SEVIRI cloud retrievals are taken to be truth. Lidar sampling errors are then simulated by sampling SEVIRI retrievals along the ground tracks of the various lidars. Lidar sampling error is measured by the difference between statistics based on SEVIRI retrievals sampled along the orbit track of each of the modeled lidars and SEVIRI retrieval statistics sampled by a hypothetical lidar with one day revisit time.

The paper is well organized and clearly written, for the most part. I have one major concern and a few specific comments.

My major concern is that Section 3.3 and Section 4 (Table 4) seem to come to opposite conclusions. Section 3.3 shows that shifting the initial day of the CALIPSO 16-day orbit cycle (essentially, shifting the orbit tracks observed on a given day) can be a major source of uncertainty. On the other hand, Table 4 shows that at the annual scale, with 10x10 grid cells, accuracy requirements can be met for most locations. Are all the results in Table 3 for 1x1 degree grid cells? Figure 5 shows that sampling uncertainties decrease when size of the grid cells increases but the uncertainties seem to be larger than what is indicated by the results in Table 4. But the metrics shown in the two sections are different and difficult to compare. Are results in the two sections consistent or do results in Table 4 ignore uncertainties due to initial day of the cycle? Please explain.

**Reply**: Thank you for your comments. In my opinion, there are two issues that need clarification.

First, Section 3.3 does indeed only consider uncertainties that resulted from the shift of the initial day of the cycle, while all other sources of uncertainty are ignored. Similarly, Section 3.1 ignores the shift of the initial day of the cycle, and only considers one particular execution of the revisit scheme (i.e., the actual calendar date of a satellite pass). The idea was to evaluate sources of uncertainty one by one. Second, Table 3 and Figure 5 give information about a range of CA/CTH, between the highest and the lowest estimates within a 16-day period (more precisely, within 16 sampling scenarios; lines 304-310). On the other hand, Table 4 gives information about how many grid cells have reported a difference that is below a threshold. Data in Table 4 were derived from differences between $n$-day and 1-day climatologies. This difference ('error') is a different measure, and is not the same as the difference between maximum and minimum values ('range') reported in Section 3.3. As the Reviewer has correctly noted, these two metrics address different aspects, and therefore are difficult to compare directly. A direct point of reference for Table 4 is Figure 1 (or Figure 2), but not Figure 5.

Following the Reviewer's comment, I have added further details, and clarification in the Discussion (see lines: 409-417).

Minor comments:

- I did not find the latitudinal extent of SEVIRI CLAAS dataset in the text. Figure 2 seems to show the CLAAS data extends from about 70S to 70N. This is important to mention in the text, to make clear that lidar sampling of the high Arctic is not being evaluated in this study.

  **Reply**: Meteosat, like any geostationary satellite, covers Earth's disc up to ~81° N/S. However, at the most extreme angles, image distortion is significant (eventually the line of sight becomes tangential to the sphere). For that reason, the study only considered Meteosat locations that were within ±70° latitude and longitude. This information has been added to the manuscript, as requested (see lines: 181-183).

- Line 133 states that Aeolus is in an equatorial (0-degree inclination) orbit. This is not correct. Aeolus is in a 97-degree inclination orbit.

  **Reply**: Thank you for pointing this out, naturally it should be 'polar' instead of 'equatorial'; now corrected.

- In Section 3.3 it is not clear what grid cell size is used in generating the statistics which are reported. Other than Figure 5, do all statistics refer to 1 degree grid boxes? What grid cell size is shown in Figure 5 c, d, g, and h?

  **Reply**: In Section 3.3, statistics have been calculated for all resolutions, but for purely technical reasons only selected results can be shown in Table 3 and Figure 5. As requested, I have added information on resolutions to the text and captions, to make it clear which data are discussed.

- Line 432: "spatial resolution above 10 degrees" is ambiguous. Does this mean "spatial resolution better than 10 degrees" ?

  **Reply**: Indeed, 'better than' is more appropriate; changed as suggested.

- Line 438. Please explain why confidence intervals are preferred over means and medians in this circumstance. Also, provide a reference on how to compute confidence intervals.

  **Reply**: Mean and median alone provide no information about the uncertainty level related to a specific sampling scheme. This information can only be represented by a confidence interval for a mean or median. If a variable follows a normal distribution, intervals can be found using the mean and standard deviation. However, in the case of cloud amount, where the distribution is frequently non-Gaussian, a (nonparametric) method like bootstrapping is much more appropriate. This information (and a reference) has been added to the manuscript, as suggested (see lines: 459-463).

---

## Author Comment (AC2)

**RC2: 'Comment on amt-2022-80', J. Yorks, 08 Jun 2022**

This paper investigates the satellite sampling of cloud amount and cloud top height from several sun-synchronous satellites hosting lidar instruments compared to SEVERI. The paper is well written, clear, and provides results that are important for designing future space-based lidar mission architectures. It deserves to be published after a few minor revisions that I believe will strengthen the paper.

My 2 major comments are:

- Diurnal variability: It is not clear to me what role diurnal variability plays in this sampling study. Based on #3 on line 156, I believe only Meteosat data from the time of the satellite overpass was used in the "truth" dataset. Is that true? If so, your monthly and annual averages are biased to the times of the overpasses and thus diurnal variability will not be accounted for. That is fine for assessing the sampling at the equatorial crossing times of each satellite, but that needs to be explicitly stated in the paper. If the SEVIRI "truth" dataset DOES include clouds from all times of day, then the geographical distribution of absolute differences in CA/CTH estimations (Figure 2) will be influenced by diurnal variability of CA or CTH. This pattern (1-day climatology was most accurately reproduced by n-day data at high latitudes but with lower accuracy at lower latitudes) would also be more consistent with where the largest diurnal variability was reported by Noel et al. (2018).

  **Reply**: The first conclusion of the Reviewer is correct. Simulations used the exact (actual) pass time of the lidar mission (a single Meteosat observation per day, per mission was used). The high (15-minute) temporal resolution of Meteosat was key, since it made it possible to select an observation that was closest in time to the lidar's overpass. Monthly/ annual means are biased to the overpass time (as in real lidar-based climatologies from sun-synchronous missions), and that was also the assumption used in this research.

  Clarification has been added, as suggested by the Reviewer (see lines: 165-167).

- Big picture impacts: The author does a great job discussing what these results mean for future lidar missions in the Conclusion. But after reading the paper, I found myself asking - what does this means for current data users? For example, if I want to use the data from these missions to compute global, annual cloud radiative effects based on CA and CTH, I can do that confidently. However, if I want to compute radiative effects at seasonal/monthly and regional/finer spatial scales, the CA and CTH from these lidar datasets may be biased based on Table 4. That is an important point for data users and will make this paper worth citing for future authors. I suggest adding a few sentences on this topic to the Discussion or Conclusion.

  **Reply**: Comments have been added to the Discussion (see lines: 423-430).

The 4 minor comments to be addressed are:

- Table 1: Do all these satellites have a 98-degree inclination angle? If not, I suggest adding an inclination angle column to this table since it impacts the repeat times.

  **Reply**: Orbital inclination has been added to Table 1, as suggested.

- Line 66: I suggest citing Yorks et al 2016 for CATS as it is more of an overview paper. The citation is: Yorks, J. E., M. J. McGill, S.P. Palm, D. L. Hlavka, P.A. Selmer, E. Nowottnick, M. A. Vaughan, S. Rodier, and W. D. Hart (2016), An Overview of the CATS Level 1 Data Products and Processing Algorithms, Geophys. Res. Let., 43, doi:10.1002/2016GL068006.

  **Reply**: The reference has been updated to Yorks *et al.* 2016. as suggested.

- Line 110-111: Are there any papers that reports the accuracy of CA and CTH from the SEVIRI products? If so, I suggest adding a sentence to report those accuracies and cite those papers. I know that doesn't impact the results of the study, but I found myself wondering what the accuracies are as I read the paper.

**Reply**: Accuracy measures have been added, as suggested (see lines 115-119).

- Line 445-446: This is a highly relevant point for future architecture designs. Did you consider looking at the ISS to provide a reference point for a lower inclination angle? I know it would be more work, but I think it would really strengthen the paper to add the ISS to this analysis. At the very least, it would be beneficial to add a sentence or two about the ISS revisit time and where it may fall compared to the satellites you studied.

  **Reply**: Yes, I did consider the ISS, but eventually decided to only base the study on polar orbiting satellites. This was because I wished to investigate lower inclination satellites separately, focusing not only on the ISS (and ISS+CALIPSO), but also the anticipated architecture(s) of the future Atmospheric Observing System (AOS). At the present moment, I have no results that would allow me to draw any conclusions regarding how the ISS compares with polar orbiters (it requires a full-scale study; I prefer not to speculate).